# Oxidative Desulfurization Activity of NIT Nitroxide Radical Modified Metallophthalocyanine

**DOI:** 10.3390/molecules27185964

**Published:** 2022-09-13

**Authors:** Min Tian, Yang He, Gai Zhang, Haibo Wang

**Affiliations:** 1School of Materials Science and Chemical Engineering, Xi’an Technological University, Xi’an 710021, China; 2Department of Chemistry, School of Pharmacy, Air Force Medical University, Xi′an 710032, China

**Keywords:** NIT nitroxide radical, metallophthalocyanine, synergistic oxidation effect, desulfurization

## Abstract

In the present study, metallophthalocyanines were modified with NIT nitroxide radicals through chemical bonds to prepare a series of metallophthalocyanines–NIT catalysts (MPcTcCl_8_-NIT, M=Mn^2+^, Fe^2+^, Co^2+^, Ni^2+^, Cu^2+^ and Zn^2+^) applied for oxidative desulfurization of thiophene (T) in model fuel. The MPcTcCl_8_-NIT catalysts were characterized by FTIR, UV-Vis, ESR, and XPS spectra. The oxidative desulfurization activity of MPcTcCl_8_-NIT catalysts was studied in a biomimetic catalytic system using molecular O_2_ as the oxidant. The MPcTcCl_8_-NIT catalysts exhibited high catalytic activities for the oxidation of thiophene in model fuel. The desulfurization rate of ZnPcTcCl_8_-NIT for thiophene reached to 99.61%, which was 20.53% higher than that of pure ZnPcTcCl_8_ (79.08%) under room temperature and natural light. The results demonstrated that MPcTcCl_8_-NIT catalysts could achieve more effective desulfurization rate under milder conditions than that of the metallophthalocyanines. The NIT nitroxide radicals also could improve the catalytic activity of metallophthalocyanine based on the synergistic oxidation effect. The stability experiments for ZnPcTcCl_8_-NIT showed that the catalyst still had a high desulfurization rate of 92.37% after five times recycling. All these findings indicate that the application of MPcTcCl_8_-NIT catalysts provides a potential new way for the desulfurization performance of thiophene in fuel.

## 1. Introduction

With people’s demand for cars increasing, the increasing types and numbers of vehicles have driven the massive using of gasoline and diesel. In recent years, vehicle exhaust has led to severe air pollution in many cities, and the air pollution index has skyrocketed. Bad weather phenomena such as dense fog, smog, and acid rain have frequently occurred. Meanwhile, the number of people with allergies and respiratory diseases has increased rapidly. All these factors have made people realize that the large consumption of gasoline and diesel has led to an irreversible impact on the atmospheric environment and human health [1,2,3,4,5,6]. This is mainly because the high sulfur content of fuel will seriously damage the catalytic converter in the exhaust pipe of an automobile, resulting in poisoning of the catalyst inside and emitting much harmful substances from the automobile exhaust. The emissions that exceed the standard limits directly endangered the atmospheric environment. With the continuous improvement and strict regulations of environmental protection laws and regulations in various countries, the sulfur content limit of vehicle fuel is becoming increasingly stringent. In view of this, it is necessary to reduce the sulfur content in gasoline and diesel. Therefore, the production of ultra-low sulfur content fuel is an inevitable choice for the national economy and people’s livelihood, as well as for environmental protection and ecology. In recent years, oxidation desulfurization technology has gradually become the research focus, and it is a promising deep desulfurization technology of fuel oil [7,8,9]. Oxidative desulfurization mainly uses oxidants to gradually oxidize sulfides into smaller substances, such as sulfones and sulfoxides, under the action of catalysts. The focus of oxidative desulfurization research is the selection and development of oxidants and catalysts.

The most commonly used oxidant for oxidative desulfurization is hydrogen peroxide, which is a strong oxidant and can be used with some homogeneous or heterogeneous catalysts to achieve deep desulfurization [10]. However, the homogeneous catalysts are difficult to be recovered which reduces the reusability of this type of catalysts [11]. The application of oxygen in the air as an oxidant for desulfurization has also become an effective method to reduce the operation cost of desulfurization [12]. However, the oxidizability of oxygen is lower than that of hydrogen oxide. It is more important to develop some new effective, economic, and pollution-free desulfurization catalysts. 

Metallophthalocyanine is a chemically stable porphyrin-like compound, that has the catalytic properties of biological enzymes and the function of activating molecular oxygen. Compared with metal porphyrins with similar structures, metallophthalocyanines have more obvious structural advantages, and the metallophthalocyanine conjugated ring structure with different substituents can be used to modify the electron cloud distribution, which is closely related to the catalytic oxidation performance, and improve the reaction activity [13]. It is expected that metallophthalocyanines can be effective desulfurization catalysts using oxygen in air as oxidant, which has attracted much attention in the field of the biomimetic catalytic oxidative desulfurization [14]. Reactive metallophthalocyanines generate highly reactive free radicals ‧OH and ‧O^2−^ by light to degrade sulfur-containing organic molecules [15]. At present, many factories at home and abroad have adopted desulfurization technology with metallophthalocyanine as a catalyst, because of its simple desulfurization technology, low cost, and high overall desulfurization efficiency [16]. However, as a catalyst, metallophthalocyanine has low catalytic activity at room temperature. Researchers found that the modification of phthalocyanine can improve its desulfurization activity [17,18].

In recent years, nitroxide radicals as oxidants have received extensive attention [19]. Nitroxide radical is an organic compound containing C, N, O, and a single spin electron. The nitroxide radicals have been successfully used in many fields such as biology, magnetism, and polymerization inhibitors due to their unique molecular space structure, single-electron delocalization, and other unique properties [20,21,22]. The stable nitroxide radical compounds are mainly divided into two classes, TEMPO (2,2,6,6-tetramethylpiperidine-1-oxyl radical) and NIT (4,4,5,5-tetramethyl-2-imidazole) oxoline-1-alkoxy-3-oxide) [23] (Figure 1a). Studies have shown that the nitroxide radicals exist in three forms: nitroxide radical, hydroxylamine and oxoammonium cations both in TEMPO and NIT, and that the redox conversion can be achieved between the three forms by gaining and losing electrons [24,25] (Figure 1b). In 1965, Golvbev found that TEMPO could be used to selectively oxidize primary alcohols to aldehydes [26], which indicated TEMPO had strong selectivity and catalytic oxidation, leading to its wide use in the field of catalytic oxidation [27,28]. For instance, TEMPO has been applied in desulfonylative α-oxyamination reactions of α-sulfonylketones [29] and fast-response electroactive actuators based on TEMPO-oxidized cellulose nanofibers [30]. However, there are few reports on the application of NIT nitroxide radicals for catalytic oxidation. Compared with TEMPO, the unpaired electrons in NIT radicals are delocalized on two equal N-O groups, resulting in a wider electron distribution and higher activity [31]. The randomness of the substituents for aldehyde group used in raw materials makes the NIT radicals a good modifying group [32,33].

In order to solve the problems that metallophthalocyanines has low catalytic activity at room temperature, the NIT nitroxide radical was bonded to the support metallophthalocyanine (MPc(COOH)_4_Cl_8_) through chemical bonds to prepare the metallophthalocyanine (MPc(COOH)_4_Cl_8_) -NIT catalysts in the present study. Using oxygen molecules in the air as the oxidant, the sulfur-containing compound thiophene was catalytically oxidized under visible light, and the cyclic regeneration performance and the stability of these new catalyst were explored by the single-variable method. The desulfurization results showed that the catalyst exhibits excellent oxidative activity for the desulfurization of fuels due to the introduction of NIT nitroxide radicals.

## 2. Results and Discussion

### 2.1. Synthesis of MPcTcCl_8_-NIT Catalysts

The synthesis process of MPcTcCl_8_-NIT catalysts was shown in Figure 2, and the specific synthesis process was illustrated by taking ZnPcTcCl_8_-NIT as an example. 

#### 2.1.1. ESR Spectra

Due to the sensitivity and accuracy, ESR spectroscopy is the best tool for the study of free radicals. As shown in Figure 3, the ESR spectra showed that the g value range of MPcTcCl_8_-NITs was from 2.0062 to 2.0071, and the ESR spectra displayed a strong signal intensity, which indicated the presence of free radicals at a high density in MPcTcCl_8_-NITs.

#### 2.1.2. IR Spectra

The FT-IR spectra measured for the MPcTcCl_8_-NIT catalysts was shown in Figure 4. The intermediate TcPcL-1 exhibited the O-H stretching vibration absorption peak at 3390 cm^−1^. The C=O stretching vibration at 1617 cm^−1^ comes from the carbonyl group formed by the connection of metallophthalocyanine complex acyl chloride and L-proline. The peaks at 2996 cm^−1^ and 2927 cm^−1^ were attributed to the C-H stretching vibration peak on the aldehyde group of the oxidized intermediate TcPcL-2, the C=O stretching vibration peak appears at 1619 cm^−1^ attributed to the carbonyl group on amide bond, and the peak at 1764 cm^−1^ was attributed to the C=O stretching vibration on the aldehyde group. It could be observed from the IR spectrum analysis that the infrared spectra for the six different central metal ions of MPcTcCl_8_-NIT compounds showed little differences from each other. The characteristic peak at 3400 cm^−1^ was obviously broadened, which was due to the stretching vibration of C-H bond caused by bonding of NIT radicals to the synthesized complex. The wavenumber of peaks ranged between 1710–1450 cm^−1^ were due to the C=C bond and C=N stretching vibration bond on the aromatic ring of metallophthalocyanine, as well as the C=O bond for acyl chlorination linking the metallophthalocyanine complex and the NIT radical moiety. The C-N single bond of the radical moiety was reflected the peak in the wavenumber range of approximately 1400–1200 cm^−1^. The C-H bending vibration peak on the phthalocyanine ring was located at 1200–1000 cm^−1^. The absorption vibration peaks at 1050–1020 cm^−1^ were attributed to the central metal ion and the M-N of the nitrogen atom on the metallophthalocyanine ring. More importantly, the characteristic peaks at approximately 790–710 cm^−1^ were attributed to the stretching vibration of the phthalocyanine ring, indicating that the phthalocyanine ring was not destroyed by the introduction of free radicals.

#### 2.1.3. UV-Vis Spectra

The UV-Vis spectra of the intermediates and MPcTcCl_8_-NIT catalysts were shown in Figure 5. From the spectrum of TcPcL-2, the peak at 278 nm indicated that the hydroxyl in TcPcL-1 was oxidized to an aldehyde group. In general, phthalocyanines showed two characteristic absorption bands in the UV-Vis spectrum. The Q absorption band between 600 and 750 nm was assigned to the transition of electrons from the highest occupied molecular orbital 6, e.g., (HOMO) to the lowest unoccupied molecular orbital 2a 1u (LUMO). The “B-band” was located at a higher p-level LUMO between 220 and 380 nm. The intensity of the Q-band of the FePcTcCl_8_-NIT and MnPcTcCl_8_-NIT catalysts decreased significantly due to the easy aggregation of the two catalysts. In addition, the aggregation effect led to the broadening of the corresponding UV absorption spectrum, and the UV absorption peak of the FePcTcCl_8_-NIT and MnPcTcCl_8_-NIT catalysts blue-shifted to 667 nm and 640 nm. The Q absorption band of ZnPcTcCl_8_-NIT underwent a redshift, first from 678 nm to 700 nm; and finally to 720 nm, due to the introduction of amino and free radical auxochromic groups onto the phthalocyanine. The intensity of the B absorption band of ZnPcTcCl_8_-NIT was enhanced. All the results indicated that the radicals were successfully immobilized onto phthalocyanine.

#### 2.1.4. XPS Spectra forMPcTcCl_8_-NIT

The XPS spectra of ZnPcTcCl_8_-NIT was shown in Figure 6a, indicating that the main elements contained in the MPcTcCl_8_-NIT catalysts were C1 s, N1 s, O1 s, and various metal elements. Since the content of the central metal ions was relatively low relative to other elements, the peaks displayed in the total element diagram also showed a low intensity. Figure 6b–e showed the high-resolution XPS spectra for each element in the ZnPcTcCl_8_-NIT radical. Figure 6b showed that the Zn 2p signals were located at approximately 1022.3 eV and 1045.6 eV, respectively, indicating that the metallophthalocyanine host was not affected by the synthesis of the catalyst. From the C1s spectrum in Figure 6c, there were three signal peaks at 284.1 eV, 286.5 eV, and 290.1 eV, and it could be observed that the carbon element in the ZnPcTcCl_8_-NIT catalyst has three bond shapes: C=N, C-N and C=O. It could be observed from Figure 6d that the signal peaks for N 1s were located at 399.8 eV and 404.9 eV, indicating that there were C-N and C=N bonds in the bonding state in the synthesized substance. As shown in Figure 6e, the O 1s peak state was the basic oxidation state of the O atom, which was displayed at approximately 532.9 eV and 537.3 eV, respectively, indicating that two forms existed: one was the C=O double bond, and the other was the N-O bond. It could be concluded that the ZnPcTcCl_8_-NIT catalyst was successfully synthesized in combination with the infrared test.

### 2.2. Desulfurization Effect of MPcTcCl_8_-NIT

The desulfurization rates of the MPcTcCl_8_-NIT catalysts with different central metal ions on thiophene model fuel were shown in Figure 7. Compared with the blank experiment (the same desulfurization conditions without catalysts) under natural illumination, the content of sulfur in model fuel was significantly decreased because of the catalytic oxidation reaction of MPcTcCl_8_-NIT catalysts. The desulfurization rates of MPcTcCl_8_-NIT catalysts with different metal centers showed some differences. The desulfurization rates of MnPcTcCl_8_-NIT, FePcTcCl_8_-NIT, CoPcTcCl_8_-NIT, NiPcTcCl_8_-NIT, CuPcTcCl_8_-NIT, and ZnPcTcCl_8_-NIT were 92.32%, 94.75%, 92.54%, 98.09%, 95.91%, and 99.61%, respectively. ZnPcTcCl_8_-NIT showed the highest desulfurization rate, reaching 99.61%. The desulfurization rates of the other MPcTcCl_8_-NIT catalysts all reached more than 90%, and all MPcTcCl_8_-NIT catalysts achieved deep desulfurization. These results indicated that the synthesized MPcTcCl_8_-NIT catalysts had a good desulfurization effect on thiophene.

As shown in Figure 8, the desulfurization reaction of pure metallophthalocyanines needed to be performed at 60 °C using a xenon lamp as a simulated light source, while for the MPcTcCl_8_-NIT catalysts, the desulfurization reactions were carried out under natural light and room temperature. The results showed that the MPcTcCl_8_-NIT catalysts could lead to a significant improvement of the catalytic degradation for the thiophene compared to the unmodified metallophthalocyanine compounds, the degree of improvement was 1.2%, 1.3%, 1.73%, 2.23%, 2.94%, and 5.53%, respectively. It was demonstrated that MPcTcCl_8_-NIT catalysts could achieve more effective desulfurization rate under milder conditions than that of the metallophthalocyanines. In order to further confirm the conclusion, the ZnPcTcCl_8_-NIT catalyst and pure ZnPcTcCl_8_ were applied to desulfurization experiments under the same conditions. The desulfurization rate of ZnPcTcCl_8_-NIT and pure ZnPcTcCl_8_ for thiophene reached 99.61% and 79.08% under the room temperature and natural light (Figure 9). Meanwhile, this result also further indicated that the influence of the NIT groups on the catalytic activity of metallophthalocyanines was obvious. The desulfurization rate of the ZnPcTcCl_8_-NIT catalyst could be increased by 20.53% due to the chemical bonding of the NIT radicals to ZnPcTcCl_8_ (Figure 9). The combination of NIT nitroxide radicals on metallophthalocyanine compounds could significantly improve the desulfurization activity for the catalyst materials.

In practical applications, the stability of these catalysts in cyclic use is an important indicator for characterizing the catalyst performance. To evaluate the cyclic stability of ZnPcTcCl_8_-NIT, a cyclic test reaction was designed. Under the optimal reaction conditions, the reusability of the catalysts with the best effect was determined. After each cyclic test of ZnPcTcCl_8_-NIT for desulfurization was complete,, ether was added to the reaction system, ZnPcTcCl_8_-NIT was precipitated, filtered, and dried under vacuum, and then the catalyst was recovered for recycling. The above operation was repeated for each loop, and the test conditions remained the same each time. The results of the recycling test of the ZnPcTcCl_8_-NIT catalyst of desulfurization for thiophene showed that under the optimal reaction conditions, the desulfurization rate of ZnPcTcCl_8_-NIT still could reach 92.37% after five times recycling, which indicated that the MPcTcCl_8_-NIT catalysts had a good recycling stability and could be reused (Figure 10).

### 2.3. Photo-Catalytic Degradation Mechanism of the MPcTcCl_8_-NIT

By summarizing the experimental results obtained for the photocatalytic performance of MPcTcCl_8_-NIT catalysts for the degradation of thiophene in n-octane, a mechanism for the oxidative catalytic desulfurization was proposed (Figure 11): metallophthalocyanine and NIT nitrogen oxygen radicals could achieve the simultaneous degradation of thiophene based on the two parts of the host. First, the thiophene molecules to be degraded were dissolved in DMF solution, so that the MPcTcCl_8_-NIT catalysts could be completely contacted with thiophene. MPcTcCl_8_-NIT was then added to the system. On one hand, the active radical components were used to catalyze the oxidation and degradation of thiophene and converted it into sulfate ions and sulfones. The oxygen in the air continued to promote the conversion of NIT nitroxide radicals from hydroxylamine to the original free radicals [34,35,36]. Meanwhile, in the presence of visible light, the main part of metallophthalocyanine absorbed photoelectrons to change its own energy and generated photoelectrons combined with oxygen [37,38,39,40]. Under the action of electron–hole pairs, O_2_-MPcTcCl_8_-NIT catalysts were immediately converted into the activated state of *O_2_-MPcTcCl_8_-NIT catalysts, and *O_2_-MPcTcCl_8_-NIT catalysts converted thiophene to sulfate ions and sulfones by oxidation. The synergistic catalytic oxidation of metallophthalocyanines and free radicals achieved the effect of deep desulfurization.

To study the role of oxygen in desulfurization, two groups of Experiments A and B were set up. For group B, air was injected into the flask with an air pump, which led to the achievement of a desulphurization rate of 99.61% after 4 h, while for group A, without oxygen, showed a desulfurization rate that remained at approximately 42% (Figure 12). From this, it could be concluded that the radical supported by the target compound was converted into hydroxylamine, which had no oxidative ability after consuming its own single electron. However, when air was continuously injected into the flask, the oxygen in the air could convert hydroxylamine into the original active free radical. As a result, the NIT nitroxide radical achieved single-electron activity recovery, continued to catalyze the oxidation of thiophene in the system, and achieved continuous desulfurization. Meanwhile, O_2_ also reacted with the phthalocyanine host under light conditions to provide holes and peroxide radicals. Therefore, in the process of phthalocyanine host and NIT nitroxide radicals catalyzing the oxidation of thiophene, the two-part synergistic catalytic oxidation of thiophene was realized (Figure 9). Finally, thiophene was converted into sulfate ions and sulfones by superoxide radicals to complete the catalytic oxidation process.

## 3. Materials and Methods

### 3.1. Characterization

In this paper, a TENSOR II infrared spectrometer (Bruker Optical Instrument Co., Ltd., Berlin, Germany) was used for testing, and the wavenumber scanning range was 400–4000 cm^−1^. The samples were prepared by pressing potassium bromide to remove moisture. The UV-Vis spectrum was tested with a UV-2550 produced by Japan’s Shimadzu Corporation. The substance to be tested was configured into a dilute solution for testing, and the ultraviolet light wavelength range was set to 200–800 nm. An Elementar Vario EL III, PE elemental analyzer was used. The EPR spectral test was used to test the ADANl SPlNSCOULD X compact electron spin resonance (ESR) spectrometer produced by ADANI company for characterization, and solid powder samples were used because of their poor solubility. X-ray photoelectron spectroscopy (XPS) was performed using an axwas ultra-spectrometer with an Al (Mono) Kα source (1486.6 eV). The thiophene concentration was determined using an Agilent 6890 gas chromatograph equipped with a flame photometric detector (FPD).

### 3.2. Synthesis Methods

All reagents were of analytical grade and used without further purification. 

#### 3.2.1. Synthesis of ZnPcTcCl_8_

Metallophthalocyanine was prepared according to the method of our previous work [16]. In total, 1.0025 g of 3,6-dichloro-1,2,4-benzenetricarboxylic acid anhydride and 0.3654 g of Zn(CH_3_COOH)_2_·4H_2_O were added as raw materials to a 100 mL three-neck flask, and then 6.0000 g of urea and 0.2500 g of NH_4_Cl were added. A mixture of 0.1200 g (NH_4_)_2_Mo_2_O_7_ was heated at 140 °C for 0.5 h with a magnetic stirrer. The reaction solution was then maintained at 220 °C for 6 h under ambient air conditions. The crude product was washed with water and 6 mol·L^−1^ HCl three times, respectively. Then, 150 mL of acetone and chloroform were used as solvents for reflux for about 12 h to obtain 1.2813 g of green solid product ZnPcTcCl_8_.

#### 3.2.2. Synthesis of the TcPcL-1 Intermediate 

The metallophthalocyanine catalyst ZnPcTcCl_8_ (1.200 g) and 50 mL dichlorosulone were mixed in a 250 mL three-necked flask and then heated to reflux for 5 h. After the solvent was removed, 100 mL dichloromethane and 6 mL trethylamine were added into the reaction system after the reaction flask was cooled in an ice water bath. Then, 3.0 g L-proline (30 mmol) and 50 mL dichloromethane mixed solution were slowly dripped into the three-neck flask, and the ice-water bath was removed after the dripping, which was stirred at room temperature for 20 h. The solvent was removed by rotary evaporation, and the intermediate 1.1 g TcPcL-1 was obtained.

#### 3.2.3. Synthesis of the TcPcL-2 Intermediate

A total of 1.0 g TcPcL-1 (4.0 mmol), 0.93 g Trichloroisocyanuric acid (TCCA), and 30 mL dichloromethane were added to a 100 mL single-necked flask in an ice-water bath, with the temperature kept at 0–5 °C. A small amount of TEMPO was added after stirring for a few minutes. As the color of the TEMPO faded, the solubility of the TCCA increased. The reaction system was stirred at room temperature for 20 min after the ice-water bath was removed. The filter cake was washed with acetone after suction filtering to obtain 0.69 g of a brown intermediate TcPcL-2.

#### 3.2.4. Synthesis of ZnPcTcCl_8_-NIT

First, 0.4 g of TcPcL-2, 0.74 g of dihydroxyamine, and 40 mL of methanol were placed into a 100 mL single-neck flask. The reactants were rapidly dissolved, and the reaction was stirred at 70 °C under reflux for 24 h. The reaction system was centrifuged at room temperature at 8000 r for 3 min and then dried in vacuum after the solvent was removed. The product was dispersed in 60 mL of secondary deionized water in an ice-water bath, and 206 mg of sodium nitrite and 3 drops of glacial acetic acid were added with stirring. The reaction was maintained at a temperature of 65 °C for 1 h. After being naturally cooled to room temperature, the final product, ZnPcTcCl_8_-NIT, was centrifuged at a speed of 8000 r for 3 min, washed twice with deionized water, and dried under vacuum.

Zn(II)PcTcCl_8_-NIT: 0.36 g, (53.86%) Yield; green solid; m.p. > 300 °C; IR (KBr) ν max/cm^−1^: 3311 (νC-H); 1642 (νC=N); 1050 (νM-N); 1311, 1167, 797 (νPc); UV-Vis (DMF) λ max/nm: 241, 296, 644, 721; Anal. Cald. for C_80_H_80_N_20_O_8_Cl_8_Zn: C, 53.42; H, 4.48; N, 15.58; Found: C, 52.85; H, 4.68; N, 15.90.

Mn(II)PcTcCl_8_-NIT: 0.24 g, (35.16%) Yield; green solid; m.p. > 300 °C; IR (KBr)ν max/cm^−1^: 3284 (νC-H); 1588 (νC=N); 1025 (νM-N); 1394, 1151, 744 (νPc); UV-Vis (DMF) λ max/nm: 269, 639, 705; Anal. Cald. for C_80_H_80_N_20_O_8_Cl_8_Mn: C, 53.73; H, 4.51; N, 15.67; Found: C, 53.86; H, 4.28; N, 15.88.

Fe(II)PcTcCl_8_-NIT: 0.27 g, (40.23%) Yield; blue solid; m.p. > 300 °C; IR (KBr) ν max/cm^−1^: 3564 (νC-H); 1622(νC=N); 1021 (νM-N); 1390, 1127, 740 (νPc); UV-Vis (DMF) λ max/nm: 234, 294, 620, 660; Anal. Cald. for C_80_H_80_N_20_O_8_Cl_8_Fe: C, 53.71; H, 4.51; N, 15.67; Found: C, 53.26; H, 4.43; N, 15.75.

Co(II)PcTcCl_8_-NIT: 0.31 g, (45.26%) Yield; blue solid; m.p. > 300 °C; IR (KBr)ν max/cm^−1^: 3457 (νC-H); 1661(νC=N); 1040 (νM-N); 1293, 1146, 750 (νPc); UV-Vis (DMF) λ max/nm: 235, 291, 361, 638, 710; Anal. Cald. for C_80_H_80_N_20_O_8_Cl_8_Co: C, 53.61; H, 4.50; N, 15.63; Found: C, 53.45; H, 4.35; N, 15.78.

Ni(II)PcTcCl_8_-NIT: 0.28 g, (39.67%) Yield; green solid; m.p. > 300 °C; IR (KBr) ν max/cm^−1^: 3331 (νC-H); 1574 (νC=N); 1031(νM-N); 1390, 1118, 779 (νPc); UV-Vis (DMF) λ max/nm:233, 269, 642, 716; Anal. Cald. for C_80_H_80_N_20_O_8_Cl_8_Ni: C, 53.62; H, 4.50; N, 15.63; Found: C, 53.55; H,4.29; N, 15.84.

Cu(II)PcTcCl_8_-NIT: 0.29 g, (42.38%) Yield; green solid; m.p. > 300 °C; IR (KBr) ν max/cm^−1^: 3564 (νC-H); 1691(νC=N); 1042 (νM-N); 1418, 1156, 787 (νPc); UV-Vis (DMF) λ max/nm: 284, 356, 644, 723; Anal. Cald. for C_80_H_80_N_20_O_8_Cl_8_Cu: C, 53.48; H,4.49; N, 15.59; Found: C, 52.85; H, 4.66; N, 15.12.

### 3.3. Evaluation of the Photocatalytic Activity

1 mL thiophene was added to 499 mL cyclohexane to prepare 500 mL 2000 µL/L model gasoline. MPcTcCl_8_-NIT catalysts (0.2 g) and model gasoline (50 mL, 2000 µL/L) were prepared in a 250 mL volumetric flask with an air pump to pass air into the flask. The reaction was carried out under room temperature, atmospheric pressure, and natural light. The sample was taken every 40 min and centrifuged at 3000 r/min for 5 min. The upper clear liquid was absorbed by the chromatograph sampler (1 µL) without the exclusion of air bubbles. To evaluate the desulfurization efficiency, each sample was measured three times by gas chromatography (Agilent 6890) to obtain the average value. The desulfurization rate was calculated by using the following formula: D (%) = (C_0_ – C_t_)/C_0_ × 100%, where D was the conversion date, C_0_ was the initial sulfur content concentration, and C_t_ was the sulfur content concentration after a period of desulfurization.

### 3.4. Evaluation of the Catalyst Recycling

The recycling process of catalysts is as follows: firstly, thiophene sulfide was oxidized to sulfoxide and sulfone by oxidant such as oxygen. Then, the diethyl ether solution was added to the reaction system and the reaction was cooled to room temperature to fully precipitate the desulfurization catalyst. Finally, the precipitated catalyst was filtered, and then the catalyst was washed with distilled water three times and dried under vacuum at 100 °C.

## 4. Conclusions

In this paper, a series of MPcTcCl_8_-NITs catalysts with different central metal ions were designed and synthesized to improve the desulfurization performance for thiophene in an n-octane-containing model fuel. Under the natural light and with oxygen in the air as oxidant, the thiophene desulfurization rate of the MPcTcCl_8_-NITs catalysts reached to 99.61% after 4 h and deep desulfurization could be achieved. The results indicated that NIT nitroxide radicals promoted the catalytic activity of metalphthalocyanine based on the synergistic oxidation effect. The stability experiments for ZnPcTcCl_8_-NIT showed that the catalysts have a good recycling stability and the desulfurization rate reached to 92.37% after five times recycling, NIT nitroxide radical modified metallophthalocyanine provided an experimental basis for the exploration and synthesis of new metal phthalocyanine desulfurization catalysts.

## Figures and Tables

**Figure 1 molecules-27-05964-f001:**
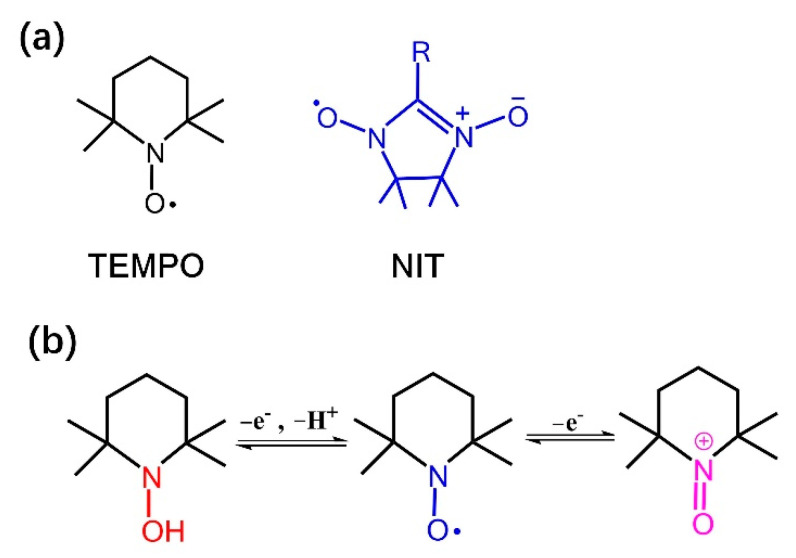
(**a**) Structure of TEMPO and NIT. (**b**) Different TEMPO oxidation states/protonation states.

**Figure 2 molecules-27-05964-f002:**
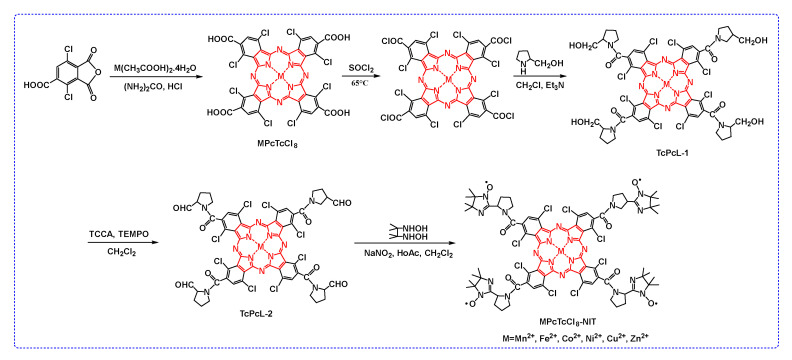
The synthesis route of MPcTcCl_8_-NIT catalysts.

**Figure 3 molecules-27-05964-f003:**
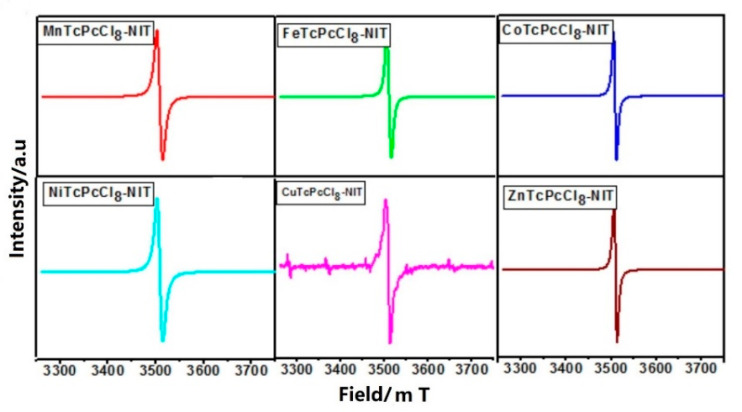
The ESR spectrum of MPcTcCl_8_-NITs.

**Figure 4 molecules-27-05964-f004:**
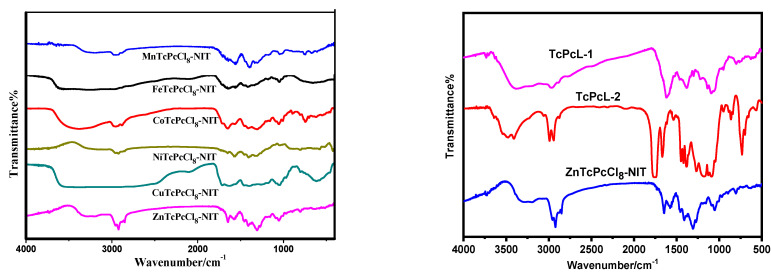
FT−IR spectra of intermediates and MPcTcCl_8_-NIT catalysts.

**Figure 5 molecules-27-05964-f005:**
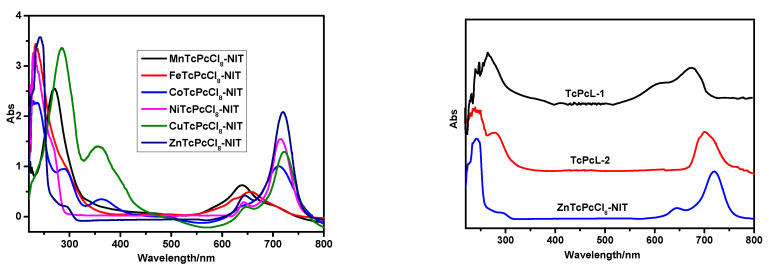
UV−Vis spectra of intermediates and MPcTcCl_8_-NIT catalysts.

**Figure 6 molecules-27-05964-f006:**
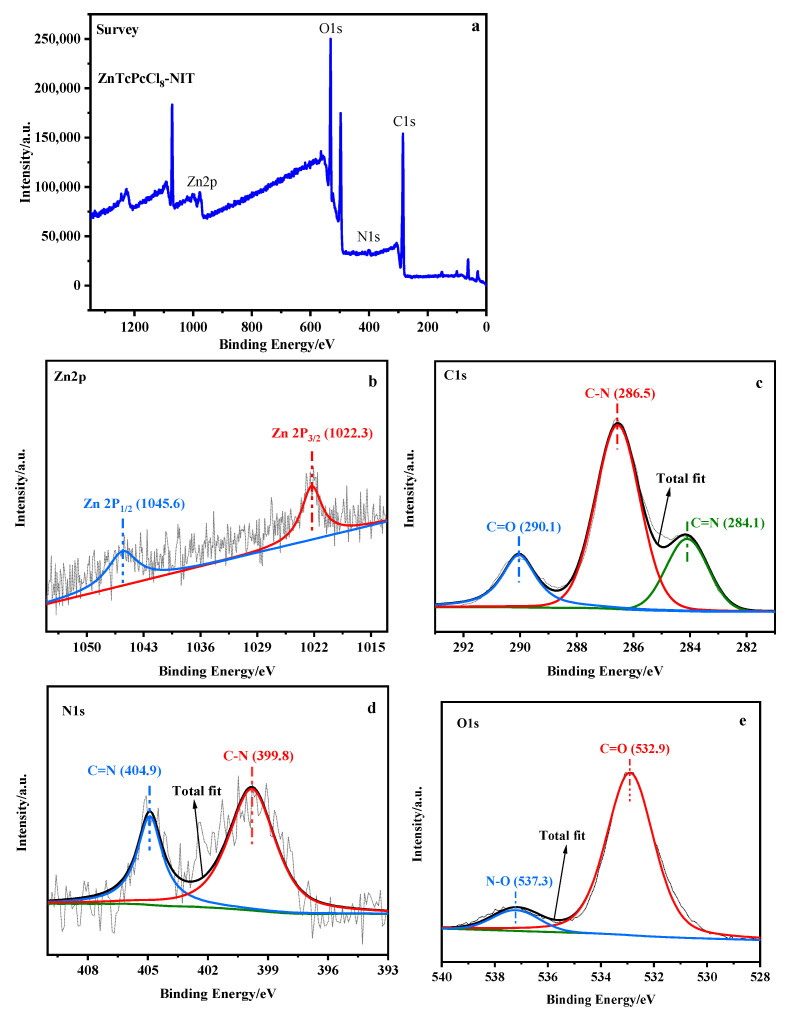
High-resolution XPS spectra of ZnPcTcCl_8_-NIT; (**a**) survey; (**b**) Zn 2p; (**c**) C1s; (**d**) N1s; (**e**) O1s.

**Figure 7 molecules-27-05964-f007:**
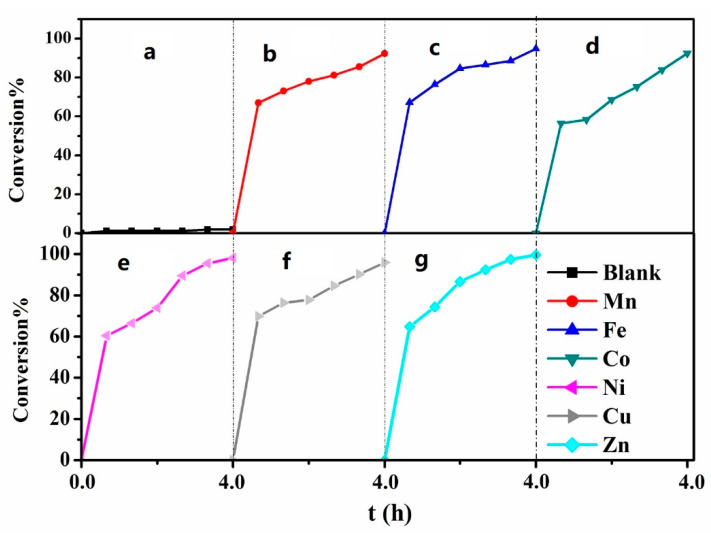
The photocatalytic desulfurization ratios of MPcTcCl_8_-NIT catalysts; (**a**) Blank sample; (**b**) MnPcTcCl_8_-NIT; (**c**) FePcTcCl_8_-NIT; (**d**) CoPcTcCl_8_-NIT; (**e**) NiPcTcCl_8_-NIT; (**f**) CuPcTcCl_8_-NIT; (**g**) ZnPcTcCl_8_-NIT.

**Figure 8 molecules-27-05964-f008:**
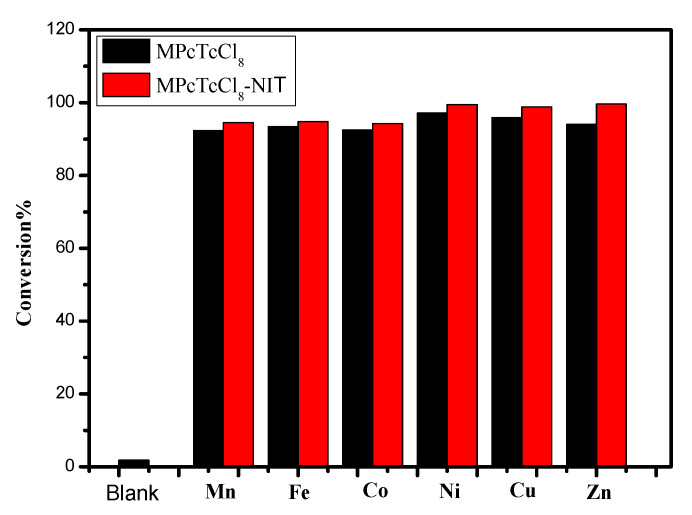
The catalytic degradation for thiophene of MPcTcCl_8_ and MPcTcCl_8_-NIT catalysts.

**Figure 9 molecules-27-05964-f009:**
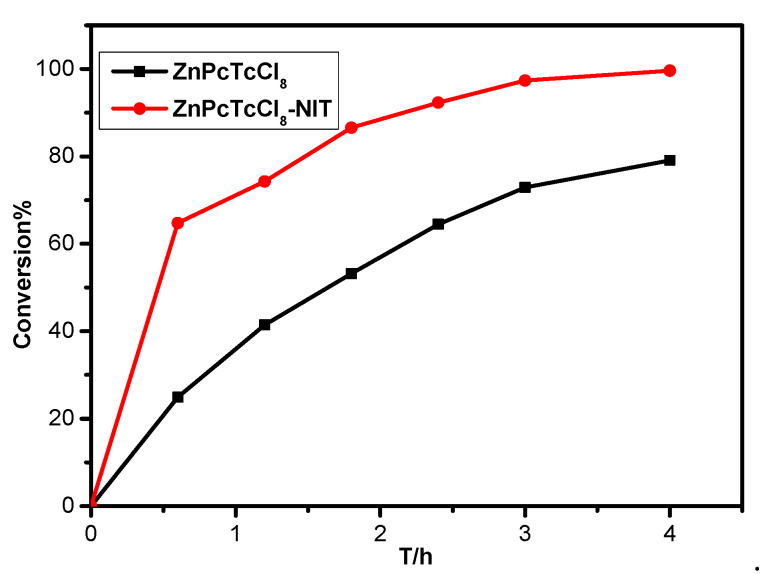
The catalytic degradation for thiophene of ZnPcTcCl_8_ and ZnPcTcCl_8_-NIT catalyst under the condition of room temperature and natural light.

**Figure 10 molecules-27-05964-f010:**
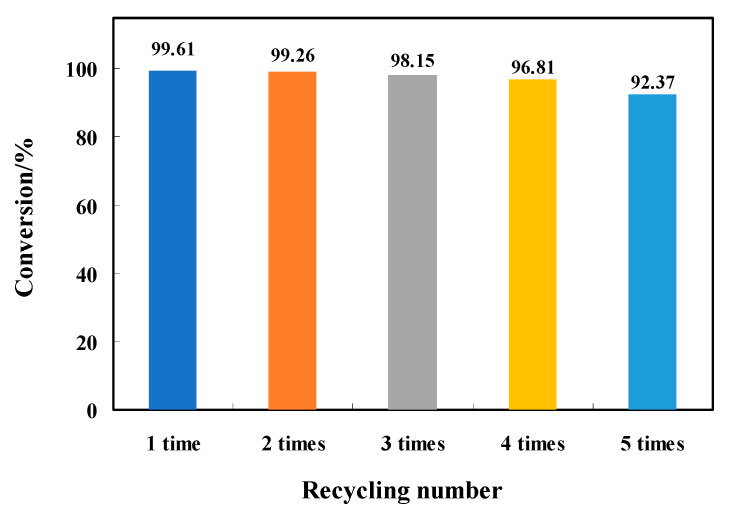
Effect of recycling times of the ZnPcTcCl_8_-NIT catalyst on desulfurization efficiency.

**Figure 11 molecules-27-05964-f011:**
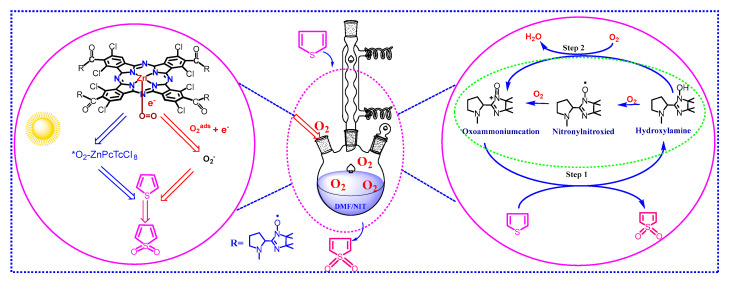
Photo−catalytic degradation mechanism of ZnPcTcCl_8_-NIT catalyst.

**Figure 12 molecules-27-05964-f012:**
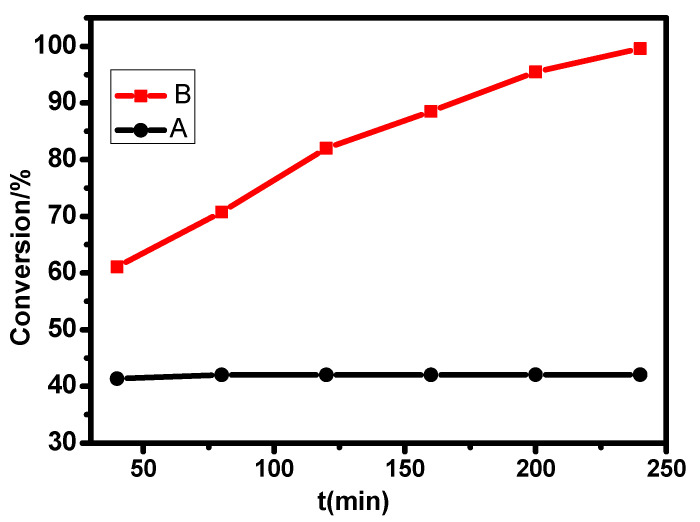
Influence of oxygen supply on desulfurization efficiency.

## Data Availability

The source data for the underlying tables and figures are available from the authors upon request.

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
