# Peer review of "Oxidative Desulfurization Activity of NIT Nitroxide Radical Modified Metallophthalocyanine"

_molecules, 2022, doi:10.3390/molecules27185964_

Round 1

Reviewer 1 Report

This manuscript by Prof. Wang et al. describes the preparation of metallophthalocyanine-NIT complexes and the study of their catalytic desulfurization performance. There are some major issues to be addressed prior to acceptance by the Journal. 

(1) In the Experimental, the metal reagents used for the synthesis need to be clearly described. 

(2) Regarding the “blank” for comparison, what is the condition for the “blank” reaction?

(3) The EPR spectrum of free NIT should be included in Fig. 3 for comparison.

(4) The content and preparation details of model gasoline are needed in the Experimental. 

(5) What does TCCA stand for? The author needs to write the full name or the chemical formula even it is commonly known.

(6) Typos are found in Fig. 4: TcPc-1, TcPc-2 vs. TcPcL-1, TcPcL-2.

(7) When the “photocatalytic desulfurization” is performed, are the dark experiments done for comparison?

(8) Compared with the desulfurization efficiency of MTcPcCl8 (>90%), the NIT substituted species improves the efficiency by 1-5.5%. If the influence of the NIT group is about up to 5%, the author may want to justify the use of NIT in the main text.

(9) What is the desulfurization efficiency for the reaction containing MTcPcCl8 + fee NIT? It should be a good comparison to see any synergistic influence of the NIT ligation.

(10) Regarding the mechanism, any data or literature results support the proposed Zn-O2*, Zn-peroxo/superoxol species?

(11) Also, the involvement of the NIT in the proposed mechanism, concerns oxygen-atom transfer. Any explanation?

Author Response

Dear Reviewer, 

Thank you very much for your comments to our manuscript. We have studied the valuable comments carefully, and tried our best to improve the manuscript and made some changes in the manuscript. We appreciate for your warm work earnestly, and hope that the correction will meet with approval.

Yours sincerely,

Min Tian

School of Materials Science and Chemical Engineering, Xi'an Technological University, Xi'an 710032, Shaanxi China

Tel.: +86 13892848785

Fax: +86 029 86173324

Email: Tianmin@st.xatu.edu.cn

Reviewer 2 Report

Review Report for “molecules-1850805”

The paper “Oxidative Desulfurization Activity of NIT nitroxide radical Modified Metallophthalocyanine” is devoted to a nitroxide radical that was bonded to the support metallophthalocyanine through chemical bonds to prepare the (MPc(COOH)4Cl8) modified by the nitroxide radical catalyst. The authors performed a detailed analysis catalytically oxidized under visible light, and the cyclic regeneration performance. The paper body is well-written. The title is interesting and involves desired novelty. However, the following major comments should be addressed:

• The abstract section needs to be more precise and clearer. It should be again revised for spelling mistakes.

• The novelty should be further strengthened. It is not clear in the current manuscript

• Explain the challenges ahead for each method in a separate section.

• Adding more results concerning UV-Vis spectra in the conclusion and results.

• Define catalyst recycling.

There are many grammatical errors; carefully check the whole manuscript.

• It was reported that in the phthalocyanine host and NIT nitroxide radicals catalyzing the oxidation of thiophene, the two-part synergistic catalytic oxidation of thiophene was realized. Prove this statement.

Author Response

Dear Reviewer,

We would like to thank you for the helpful comments and suggestions. We have studied the valuable comments carefully, and tried our best to improve the manuscript and made some changes in the manuscript. We appreciate for your warm work earnestly, and hope that the correction will meet with approval.

Yours sincerely,

Min Tian

School of Materials Science and Chemical Engineering, Xi'an Technological University, Xi'an 710032, Shaanxi China

Tel.: +86 13892848785

Fax: +86 029 86173324

Email: Tianmin@st.xatu.edu.cn

Round 2

Reviewer 1 Report

The author has made corresponding changes according to reviewer's comments. It is suitable for the Journal.

Reviewer 2 Report

The authors revised the manuscripts based on the reviewer's comments.